# Protective Effects of Sitagliptin on Streptozotocin-Induced Hepatic Injury in Diabetic Rats: A Possible Mechanisms

**DOI:** 10.3390/diseases11040184

**Published:** 2023-12-18

**Authors:** Qamraa H. Alqahtani, Samiyah Alshehri, Ahlam M. Alhusaini, Wedad S. Sarawi, Sana S. Alqarni, Raessa Mohamed, Meha N. Kumar, Juman Al-Saab, Iman H. Hasan

**Affiliations:** 1Department of Pharmacology and Toxicology, College of Pharmacy, King Saud University, P.O. Box 22452, Riyadh 11495, Saudi Arabia; ghamad@ksu.edu.sa (Q.H.A.); saalshehri@ksu.edu.sa (S.A.); aelhusaini@ksu.edu.sa (A.M.A.); wsarawi@ksu.edu.sa (W.S.S.); 443203447@student.ksu.edu.sa (J.A.-S.); 2Department of Clinical Laboratory Science, College of Applied Medical Sciences, King Saud University, P.O. Box 2925, Riyadh 11461, Saudi Arabia; saalqarni@ksu.edu.sa; 3Department of Histology, College of Medicine, King Saud University, P.O. Box 2925, Riyadh 11461, Saudi Arabia; rmohammad@ksu.edu.sa; 4Department of Clinical Medicine, Shanghai Medical College, Fudan University, Shanghai 200233, China; 17301056073@fudan.edu.cn

**Keywords:** diabetic, dipeptidyl peptidase-4 inhibitor, sitagliptin, STZ, mTOR/NF-κB/NLRP3 signaling

## Abstract

Diabetes is a ubiquitous disease that causes several complications. It is associated with insulin resistance, which affects the metabolism of proteins, carbohydrates, and fats and triggers liver diseases such as fatty liver disease, steatohepatitis, fibrosis, and cirrhosis. Despite the effectiveness of Sitagliptin (ST) as an antidiabetic drug, its role in diabetes-induced liver injury is yet to be fully investigated. Therefore, this study aims to investigate the effect of ST on hepatic oxidative injury, inflammation, apoptosis, and the mTOR/NF-κB/NLRP3 signaling pathway in streptozotocin (STZ)-induced liver injury. Rats were allocated into four groups: two nondiabetic groups, control rats and ST rats (100 mg/kg), and two diabetic groups induced by STZ, and they received either normal saline or ST for 90 days. Diabetic rats showed significant hyperglycemia, hyperlipidemia, and elevation in liver enzymes. After STZ induction, the results revealed remarkable increases in hepatic oxidative stress, inflammation, and hepatocyte degeneration. In addition, STZ upregulated the immunoreactivity of NF-κB/p65, NLRP3, and mTOR but downregulated IKB-α in liver tissue. The use of ST mitigated metabolic and hepatic changes induced by STZ; it also reduced oxidative stress, inflammation, and hepatocyte degeneration. The normal expression of NF-κB/p65, NLRP3, mTOR, and IKB-α were restored with ST treatment. Based on that, our study revealed for the first time the hepatoprotective effect of ST that is mediated by controlling inflammation, oxidative stress, and mTOR/NF-κB/NLRP3 signaling.

## 1. Introduction

Diabetes mellitus (DM) is a persistent metabolic condition characterized by a decrease in insulin secretion or action [1]. In 2019, the estimated prevalence of DM in the world was 9.3% (463 million individuals), and by 2030, it is predicted to be increased to 10.2% (578 million) [2]. In Saudi Arabia, the prevalence of Type 2 diabetes mellitus (T2DM) is 16.4%, with the potential that the prevalence will increase [3]. In DM, hyperglycemia is associated with metabolic dysfunctions of carbohydrates, protein, and fat that affect different organs by disrupting their normal functions. Long-term hyperglycemia progresses gradually and arises in both microvascular and macrovascular complications that affect multiple organs including the liver [4].

Recent research revealed a strong link between DM and numerous liver abnormalities, including hepatocellular damage in non-alcoholic fatty liver disease (NAFLD), cirrhosis, and viral hepatitis [5,6]. Oxidative stress plays a crucial role in the activation of some genes involved in inflammatory pathways that can cause many chronic disorders. Nevertheless, oxidative stress can be initiated by the excessive release of pro-inflammatory cytokines including interleukin (IL-6), interleukin (IL)-1β, and tumor necrosis factor-α (TNFα). It has been shown that such cytokines can increase the accumulation of oxidative damage products in the liver, such as fluorescent pigments, malondialdehyde (MDA), and conjugated dienes [7,8].

Nuclear factor-kappa B (NF-κB) is crucial for innate immunity and inflammation regulation, and it is required for pro-inflammatory cytokine expression, such as IL-1β, IL-6, and IL-18. The expression of NF-kB is highly regulated by making a complex with inhibitor of NF-κB (IκB) proteins. This complex blocks the binding ability of NF-κB to DNA, so the complex is exported to the cytoplasm [9]. A previous study suggested that NF-κB activation is a crucial event early in the pathobiology of diabetes [10]. Furthermore, NF-κB p65 inactivation in hepatocytes improves hepatic insulin sensitivity and downregulates cAMP/PKA downregulates [11]. Activation of the NF-κB pathway causes activation of the NOD-like receptor protein 3 (NLRP3) inflammasome, which is a multiprotein complex that affects the development of liver disorders such as viral fulminant hepatitis, alcoholic steatohepatitis, and non-alcoholic fatty liver disease through processing caspase-1 cleavage and IL-1β secretion [12,13,14]. The NLRP3 inflammasome can be activated by excess ATP, uric acid, and mitochondrial DNA, as the mitochondria-derived reactive oxygen species (mtROS) is a significant activator [15,16].

Additionally, in hepatocytes, a recent study showed a significant effect of exogenous hydrogen sulfate in inhibiting NLRP3 inflammasome-mediated inflammation via promoting autophagy through the AMPK/mTOR pathway [17]. Over the past decade, the mechanistic target of rapamycin (mTOR) has become an essential regulator of cell growth and metabolism. Deregulation of the mTOR signaling system is connected to aging and the development of disease [18,19]. Caspases are a family of protease enzymes crucial for programmed cell death, and inflammation involves proteolytic caspases in apoptosis activation and execution [20,21]. Apoptosis stimulates the initiation of the activation of caspases (caspase-2, -8, -9, and -10), which cleave and thereby activate the effector members (caspase-3, -6, and -7) [22]. In the early stage of type 2 DM, activation of caspase-3 plays an essential role in disease development and onset through decreasing beta-cell mass due to apoptosis [23].

Sitagliptin (ST) is a dipeptidyl peptidase-4 inhibitor (DPP4) used to control blood glucose levels. DPPA4 is a glycoprotein enzyme expressed in most cell types; it is responsible for the degradation of incretin hormones such as glucagon-like peptide-1 (GLP-1). GLP-1 is a hormone released from L-cells in the intestine that increases insulin secretion to regulate the circulating level of glucose. Ghorpade et al. reported that the secretion of DPP4 from hepatocytes triggered adipose tissue inflammation and insulin resistance, while silencing of DPP4 reduced them; however, this effect was not produced with ST, indicating a diverse mode of action [24] Previous studies reported the hepatoprotective activity of ST in a mouse model of steatohepatitis via modulation of oxidative stress, lipid metabolism, and inflammatory mediators [25,26]. Besides that, new evidence suggested the protective effect of ST through the modulation of Nrf2 and NF-κB signaling pathways with subsequent suppression of inflammatory and apoptotic processes in methotrexate toxicity [27].

Due to the multifactorial aspects of liver injury in diabetic patients, the exact mechanism of fibrosis formation is not fully understood. Thus, the aims of the present study are to investigate the link between hepatic expression of NF-κB, caspase-1, and the NLRP3 inflammasome and diabetes, and to assess the hepatoprotective effect of ST via measuring the expression levels of oxidative stress and inflammatory biomarkers in a diabetic rat model.

## 2. Materials and Methods

### 2.1. Chemicals and Reagents

The drug Sitagliptin (ST), chemicals to induce diabetes (streptozotocin (STZ)), chemical compounds to measure reduced glutathione (GSH), and malondialdehyde (MDA) were supplied by Sigma–Aldrich Comp. (3050 Spruce, St. Louis, MO, USA). On the other hand, the spectrophotometric kits for total cholesterol, high-density lipoprotein (HDL), and triglycerides were obtained from Accurex (Mumbai, India). Antibodies were obtained for NLRP3 (Cat. Number MA5-32255), NF-κB/p65 (Cat. number ab16502), IKB-α (Cat. Number ab-76429), and mTOR (Cat. number ab32028).

### 2.2. Induction of Diabetes in Rats

Type 2 DM was induced in rats by administering a single intraperitoneal (i.p.) injection of 55 mg/kg body weight of streptozotocin (STZ) after an overnight fast [25]. The STZ solution was freshly prepared by dissolving it in 0.1 M cold citrate buffer with a pH of 4.5. Diabetes was confirmed by measuring the rats’ blood glucose levels 72 h (3 days) after the STZ injection using a MEDISAFE MINI blood glucose reader made by the TERUMO Corporation in Tokyo, Japan. Rats with blood glucose levels > 200 mg/dL were considered diabetic and used for subsequent experiments. To ensure the persistence of diabetes, blood glucose levels were measured again 7 days after the STZ injection.

### 2.3. Animals and Experimental Design

Male Wistar albino rats weighing between 180 and 200 g were included in this investigation. The animals were purchased from the Bio-Resource Unit at King Saud University’s College of Pharmacy. They were fed a regular rat pellet diet and unlimited water.

A total of 32 rats were used in the experiment and divided into four groups, each consisting of 8 rats as follows:

Group I (control): Nondiabetic rats received physiological saline orally for 90 days.

Group II (ST): Nondiabetic rats received ST (100 mg/kg) [26] dissolved in saline by oral gavage for 90 days.

Group III (diabetic): Diabetic rats received physiological saline by oral gavage for 90 days.

Group IV (diabetic + ST): Diabetic rats received sitagliptin (100 mg/kg) for 90 days.

At the end of the experiment, all rats were euthanized using CO_2_ and killed by decapitation. Blood samples were collected for the sera separation for measuring lipid and aminotransferase enzymes. Liver tissues were excised, weighed, and divided into several sections for various analyses: The first sections were homogenized in phosphate-buffered saline for biochemical analysis of antioxidant and inflammatory biomarkers. The second sections of liver samples were fixed in 10% formalin for histopathological and immunohistochemical examinations.

### 2.4. Biological and Biochemical Analysis

#### 2.4.1. Calculating the Change in Body Weight and Liver Weight/Body Weight Ratio (LW/BW)

The weights of the rats in all groups were measured at the beginning of the experiment and before sacrifice. Also, the livers’ weights were measured after dissection (liver weight, LW) for all groups, and the LW/BW ratio was calculated as follows:LW/BW ratio = liver weight (g)/body weight (g) × 100.

#### 2.4.2. Assay of Serum Lipids and Hepatic Function Markers

The serum levels of total cholesterol (TCH, Cat. No. CH200) and high-density lipoprotein (HDL, Cat. No. CH1383) cholesterol were measured using commercially available kits (Randox Laboratories Ltd., Kearneysville, WV, USA) while serum levels of triglycerides (TG, Cat. No. 2200-430) were measured using a colorimetric kit (EKF Diagnostics Inc., Boerne, TX, USA).

For liver enzymes, the levels of serum alanine aminotransferase (ALT, AL1205) and aspartate aminotransferase (AST, AS3804) were evaluated using a colorimetric assay from Randox Laboratories Ltd. (USA), based on the instructions provided by the manufacturer.

#### 2.4.3. Assay of Lipid Peroxidation, GSH, and SOD

MDA, an index of lipid peroxidation, GSH, and SOD were assayed in the liver homogenates according to the methods described previously in [27,28,29] and [30], respectively.

#### 2.4.4. Determination of Inflammatory Mediators

Hepatic tumor necrosis factor-α (TNF-α), interleukin 1-β (IL-1β), and interleukin 18 (IL-18) were measured using ELISA kits, and the procedures were applied according to the manufacturer’s instructions.

### 2.5. Histopathology and Immunohistochemistry

Immediately after sacrifice, the livers were excised and washed to remove any external contaminants or blood. The washed liver tissue was then fixed in 10% neutral buffered formalin for 24 h. After fixation, the liver tissue is processed for sectioning and then embedded in paraffin wax. Five-micrometer (μm)-thick sections were cut from the embedded liver blocks. Standard hematoxylin and eosin (H&E) were used for staining after tissue dewaxing and rehydration. This staining method allows the visualization of tissue structure and cell morphology and identifies any abnormalities or changes in the liver tissue.

Other sections of the liver were used for immunohistochemistry to detect the immunoreactivity of mTOR, NF-κB/p65, IKB-α, and NLRP3. After tissue dewaxing and rehydration, the liver sections were incubated in a 0.3% hydrogen peroxide solution to quench endogenous peroxidases. After that, the sections were washed in tris-buffered saline (TBS; pH 7.6). This step is important to remove any residual hydrogen peroxide and prepare the tissue for antibody incubation. The slides were incubated for one hour with Novocastra Protein Block (RE7102, Leica Biosystems, IL, USA) to prevent the non-specific binding of antibodies to the tissue. Subsequently, the liver tissue sections were probed with specific primary antibodies (mTOR, NF-κB/p65, IKB-α, NLRP3) that were diluted to the manufacturer-recommended concentration and incubated at 4 °C overnight. On the next day and after extensive washing, the sections were probed with biotinylated secondary antibody for two hours at room temperature. The signal was amplified by incubating the section with an avidin–biotin–peroxidase complex (ABC, Vector Laboratories, Newark, CA, USA) for 30 min. The chromogenic substrate diaminobenzidine (DAB, Vector Laboratories, USA) was added to the slide until a brown precipitate appeared and was then counterstained with hematoxylin. Negative control sections were simultaneously stained but with omitted primary antibody incubation.

### 2.6. Statistical Analysis

The results of our study were obtained as mean ± standard error (SEM). All statistical comparisons were carried out by using GraphPad Prism 9 (GraphPad Software, San Diego, CA, USA) and the one-way ANOVA test was applied followed by Tukey’s post hoc test; significance was defined as *(p* < 0.05).

## 3. Results

### 3.1. Sitagliptin Attenuates Body Weight Loss and Hyperglycemia and Reverses Elevated Liver Function Biomarkers in Diabetic Rats

Diabetic rats showed a significant (*p* < 0.001) body weight loss when compared with the control group. ST significantly (*p* < 0.001) attenuated body weight loss in diabetic rats when supplemented for 90 days. For 90 days, normal rats that received ST showed nonsignificant (*p* > 0.05) changes in body weight when compared with the control group (Figure 1A).

Our results (Figure 1B) showed a significant (*p* < 0.001) increase in blood glucose levels when compared with the control group, as well as a significant (*p* < 0.001) reduction in blood glucose levels after ST administration in treated rats when compared with the diabetic rats. The supplementation of ST for 90 days did not affect blood glucose levels in normal rats. As shown in Figure 1C, LW/BW was significantly (*p* < 0.001) elevated in the diabetic group when compared with the control group, an effect that was repressed after ST administration. In addition, ST treatment for 90 days did not affect the LW/BW of normal rats, as represented in Figure 1C.

To investigate hyperglycemia-induced hepatic injury and the possible protective role of ST, we determined serum levels of ALT and AST and performed a histological analysis. Diabetic rats revealed upregulation (*p* < 0.001) in serum ALT and AST compared with the nondiabetic group. In contrast, diabetic rats treated with ST exhibited a marked (*p* < 0.001) improvement in ALT and AST compared with the nontreated group (Figure 2). ST treatment of normal rats did not induce significant changes in either ALT or AST compared to control rats.

### 3.2. Sitagliptin Attenuates Serum Lipid Profile in Diabetic Rats

Data summarized in Table 1 represent the effect of ST on the serum lipid profile and hepatic risk induced in STZ-induced diabetic rats. Diabetic rats exhibited a significant (*p* < 0.001) increase in serum TG and TCH, while the significant reduction in serum levels of HDL cholesterol was clear when compared with the corresponding nondiabetic group. On the other hand, diabetic rats treated with ST showed significant (*p* < 0.001) beneficial modulation in all lipid profile markers. Nondiabetic rats treated with ST showed no change in their lipid profile when compared with the control group.

### 3.3. Sitagliptin Suppresses Diabetes-Induced Oxidative Stress in The Liver Tissues of Diabetic Rats

Diabetic animals exhibited a remarkable (*p* < 0.001) increase in the levels of hepatic MDA compared with normal ones (Figure 3A). Treatment with ST significantly (*p* < 0.001) decreased hepatic levels of MDA in diabetic rats. In normal rats, ST administration for 90 days did not affect MDA levels.

In addition, diabetic rats showed a significant decrease (*p* < 0.001) in hepatic SOD activity (Figure 3B) and GSH levels (Figure 3C) when compared with the normal nondiabetic group. The use of ST in diabetic rats for 90 days significantly ameliorated the reduction in SOD (*p* < 0.01) activity and GSH content (*p* < 0.001) and in the hepatic tissue while exerting a nonsignificant (*p* > 0.05) effect in normal rats.

### 3.4. Sitagliptin Improves Histological Architecture and Decreases Collagen Deposition in the Liver Tissues of Diabetic Rats

Histopathological examination of the H&E-stained liver sections of nondiabetic rats showed normal liver tissue with normal features of the architecture of hepatic lobules and portal tracts. (Figure 4A,B). Liver sections from diabetic rats revealed multiple patches of hepatic degeneration with the loss of a group of hepatocytes, inflammatory cellular infiltrations, and hepatocyte cytoplasmic degeneration (Figure 4C). On the other hand, sections from the liver of diabetic rats receiving ST showed a marked decrease in inflammatory cellular infiltrations; most of the hepatocytes were apparently with normal cytoplasm and nuclei (Figure 4D).

### 3.5. Sitagliptin Inhibits Inflammation and Hepatic Apoptosis in Diabetic Rats

The ameliorative effect of ST on diabetes-associated inflammation was explored via the determination of hepatic IL-1β, IL-18, and TNF-α levels. These inflammatory mediators were significantly (*p* < 0.001) increased in diabetic rats (Figure 5A–C) respectively. ST significantly decreased TNF-α, IL-1β, and IL-18 in hepatic tissues of the STZ-induced diabetic rats. Oral administration of ST to normal rats exerted nonsignificant (*p* > 0.05) effects on hepatic IL-1β, IL-18, and TNF-α.

To investigate the protective effect of ST on hepatic apoptosis in the diabetic rat model, the expression of caspase-1 was assessed by ELISA (Figure 5D). STZ-induced diabetic rats showed a marked increase in caspase-1 levels in hepatocytes, an effect that was significantly abolished by ST. Treatment of normal rats with ST did not induce significant changes in caspase-1 levels.

### 3.6. Sitagliptin Regulates Protein Expression of NF-κB/p65, IKB-α, NLRP3, and mTOR Cascades in the Hepatocytes of Diabetic Rats

The effect of ST on the expression of NF-κB/p65 and IKB in the hepatic tissue of normal and diabetic rats was determined by immunohistochemical staining. Liver sections from nondiabetic control and ST-treated rats revealed very faint immune reactive signals of NF-κB/p65 in the nuclei and cytoplasm (Figure 6A,B). Liver sections of the STZ-induced diabetic rats showed strong immunoreactivity of hepatocyte nuclei and a moderate signal in the cytoplasm (Figure 6C). In contrast, the diabetic rats treated with ST showed few immunopositivity nuclei (Figure 6D). Similarly, the immunohistochemical staining of the liver with IKB-α revealed high and moderate immune-positive signals, especially in the nuclei of normal control (Figure 6E) and ST-treated rats (Figure 6F), respectively. STZ-induced diabetic rats showed low immune reactivity of almost all hepatic cell nuclei (Figure 6G). On the other hand, diabetic rats treated with ST for 90 days showed many immunopositivity nuclei of hepatic cells (Figure 6H).

On the other hand, NLRP3 immune-stained liver sections (Figure 7A,B) of a normal control liver and liver of a rat that received ST, respectively, showed a normal absence of nuclear immune reactivity, while liver sections from a diabetic rat showed many hepatocytes with immune-stained nuclei (Figure 7C), while sections from an ST-treated rat (Figure 7D) showed a marked decrease in the hepatocytes’ nuclear immune reactivity.

Diffuse overexpression of mTOR was observed in almost all liver sections from diabetic rats (Figure 8C), while no positive cytoplasmic immunostaining was observed in nondiabetic rats (Figure 8A,B). However, Figure 8D showed significant inhibition in the immune passivity of mTOR expression after ST treatment.

## 4. Discussion

DM is the most prevalent metabolic disease characterized by impairment in insulin release, insulin resistance, and hyperglycemia. Long-term hyperglycemia can cause microvascular and macrovascular complications that affect different organs, including the liver [31]. However, due to the multifactorial aspects of liver injury in diabetic patients, the exact mechanism of metabolic and pathological changes is not fully understood. ST belongs to the new generation of oral antidiabetics and showed a hepatoprotection mechanism [32,33]. The present study aimed to investigate the possible mechanisms of hepatoprotection of ST in DM, pointing to the involvement of the mTOR/NF-κB/NLRP3 signaling pathway.

Indeed, few studies have found a link between weight loss after diabetes diagnosis and its complications. In the present study, the T2DM rat model showed a marked reduction in BW at the end of the experiments compared to the final BW of nondiabetic control groups. In addition, the blood glucose levels and the liver to body weight ratio increased compared to nondiabetic controls. This result is similar to the finding of a study conducted on patients by Yang et al. [34]. To the best of our knowledge, no study has been conducted to assess the changes in BW and LW in T2DM rat models. The administration of ST can ameliorate STZ’s effect on the liver by restoring normal blood glucose levels, BW, and LW. Also, there is a significant reduction in lipid profiles (TG, TCH) and an increase in HDL. Likewise, an open-label observational study was conducted on adult diabetic patients and found that ST improves blood glucose control, body weight, blood pressure, and lipid profile [35]. Nonetheless, no previous study has linked ST’s effect on the liver and body weight.

T2DM is strongly associated with oxidative stress production and is mainly manifested by an increased generation of reactive oxygen species (ROS). This leads to the damage of proteins, lipids, nucleic acids, and intracellular organelles, such as mitochondria. Furthermore, ROS contribute to the progressive failure of pancreatic β-cells and the impairment of insulin action in target tissues, especially skeletal muscle, liver, and adipose tissue [36]. The present study showed a similar effect in the STZ rat model, which resulted in oxidative stress, as evidenced by elevated liver enzyme levels with reduced SOD activity and GSH, and increased MDA levels compared with the control animals. MDA is one of the critical markers related to ROS and oxidative damage, and when free radicals increase, MDA production will be increased [37]. Furthermore, SOD is an important antioxidant enzymatic found in most organs, including the liver; decreased SOD enzyme levels occur with cell damage [38]. Finally, the essential role of GSH content in regulating hepatocyte cell death was documented, where an increase in ROS resulted in the depletion of GSH [39].

Our findings indicate that the administration of ST decreased MDA levels and increased SOD; these results support Nader et al.’s study [40]. Similar antioxidant effects of ST were recently reported in many inflammatory conditions, such as liver steatohepatitis [41] and T2DM [42]. Besides the liver, ST can attenuate STZ-induced oxidative stress and glomerular lesions by decreasing DPP-4 protein levels and upregulating GLP-1 in renal tissue [2]. Moreover, the expression of DPP-4 in hepatocytes of patients with hepatitis C virus (HCV) infection was found to be high [43]. In addition, proinflammatory cytokine expression and the extent of hepatic steatosis were lower in the livers of DPP-4-deficient rats compared to wild-type rats [44]. Therefore, by inhibiting DPP-4, sitagliptin allows GLP-1 to remain active for a longer duration. Indeed, GLP-1 has been shown to have protective effects on the liver, including promoting hepatocyte survival and reducing inflammation [45]. Furthermore, GLP-1 receptor agonists were found to significantly reduce hepatic cell inflammation, necrosis, and apoptosis [46].

In the scope of this study, the AST and ALT level was considered the initial indicator for liver injury [47]. The results revealed the release of liver enzymes such as ALT and AST, which were increased in T2DM rats compared to the normal control group and reduced with ST treatment. ST has a protective effect on the liver, manifested as an improvement in liver transaminases [48]. These results were supported by histopathological examination findings, which disclosed marked hepatic injury in the T2DM group. The biochemical and histological alterations associated with STZ induction were significantly decreased by ST treatment, indicating that ST could effectively counteract DM-induced liver cell injury.

Besides oxidative stress, the inflammatory processes were implicated in the onset of DM. The interplay between oxidative stress and inflammation has recently been well established in this disease. Inflammatory disease plays a significant role in insulin resistance and T2DM progression. The main sites of inflammation include adipose tissue, the liver, skeletal muscle, and the pancreas [40,49]. Previous studies demonstrate a link between inflammatory processes and DM, especially in abdominal obesity, associated with low-grade systemic inflammation [50]. TNF-α is a pro-inflammatory cytokine necessary in mediating the inflammatory response [51]. IL-6 is involved in the acute hepatic inflammation phase and is produced instantly and transiently in infection response [52]. In immune cells, caspase-1 cleaves pro-IL-1β and pro-IL-18, converting them into active forms in response to inflammatory stimuli [53]. Our results showed that the hepatic tissues of the diabatic group have increased expression and release of caspase 1, TNF-α, IL1-β, and IL-18. In contrast, T2DM groups with ST showed significant downregulation of these elevated inflammatory markers, which agrees with previous studies on ST’s ability to counteract the lipopolysaccharide-induced release of various cytokines [54]. As far as we know, this is the first study to find that ST induces a significant reduction in IL18 expression in rat hepatocytes.

The NLRP3 inflammasome is essential in chronic low-grade metabolic inflammation, and its excessive activation may contribute to the beginning and progression of disease. This complex includes an effector (caspase-1), an adaptor (apoptosis-associated speck-like protein, or PYCARD), and a sensor (NLRP3). When activated, pro-caspase-1 oligomerizes, leading to the production of the pro-inflammatory cytokines IL-1β and IL-18. The NLRP3 inflammasome is activated by different factors, including pro-inflammatory cytokines, inflammatory signals, TNF, and high glucose [55]. Our finding demonstrates that ST induced a marked decrease in NLRP3 hepatocyte nuclear expression, which was upregulated in DM. These results are in parallel with the results of the ST against radiation-induced intestinal injury study that found that ST enormously decreased the expressions of NLRP3, caspase-1, and IL-1β [56].

ST showed hepatoprotective effects against thioacetamide-induced liver injury by downregulating NLPR3 inflammasome production and TLR4 and NF-κB protein expression [44]. In addition, ST was effective against inflammation, fibrosis, and tubulointerstitial injury in doxorubicin-induced nephropathy. The gene expressions of NLRP3, caspase-1, and IL-1β were elevated during doxorubicin use but reversed with ST treatment [45]. The inhibitory effect of ST on these proteins can be mediated through protein kinase C inhibition [46]. However, to the best of our knowledge, there was no study that reported the effect of ST on the hepatic NLRP3, caspase-1, and IL-1β in diabetes rat models induced by STZ, which emphasizes the importance of this study.

Previously, it was reported that NF-κB activation leads to the induction of cytotoxic cytokines, which aggravate liver damage [57]. However, the NF-κB/Rel family comprises five members, including p65 (Rel-A) proteins. The most abundant form of NF-κB activated by pathologic stimuli through the canonical pathway is the p65:p50 heterodimer. The development of many chronic diseases depends on an abnormal increase in activated p65 and the consequent transactivation of effector molecules [58]. Our results, in agreement with previous studies, demonstrate that T2DM strongly increases NF-κB/p65 expression, whereas ST treatment decreases this effect. ST treatment showed a marked inhibitory effect on the NF-κB activation pathway. Previous reports supported these results by demonstrating the ability of this drug to counteract NF-κB activation [59] and ST hepatoprotection and its ability to modulate NF-κB and Nrf2 crosstalk [33].

Recently, autophagy has emerged as a pertinent underlying mechanism for the pathogenesis of diabetes. Long-term exposure to hyperglycemia induces apoptosis and dysfunction of islet β cells [60]. There are several signaling pathway-dependent regulation processes of autophagy, including mTOR, which is necessary to operate several of these pathways [61]. Our immunohistochemistry result showed that ST decreased mTOR in the hepatocytes of diabetic rats. There are several studies confirming the implication of mTOR expression in the protective activity of ST in different organs’ injuries. A recent study on acute pancreatitis-related acute lung injury found that ST alleviated oxidative stress and excessive autophagy [62]. In polycystic ovary syndrome (PCOS) rat models, ST inhibited autophagy and inflammation by inhibiting the PI3K/AKT/mTOR and TLR4/NF-κB pathways, respectively [63]. Furthermore, the in vivo reports showed that ST downregulated the expression of mTOR protein in rat testes, thus modulating autophagy [64].

## 5. Conclusions

These results corroborate emerging evidence for the damaging effect of STZ on liver tissue by inducing oxidative stress, inflammation, and apoptosis. Treatment with ST prevented STZ-induced hepatic injury by reversing its metabolic, inflammatory, apoptotic, and pathological changes in hepatic tissue. Further, ST restored the normal expression of effects against STZ-induced liver injury by inhibiting hepatic inflammation, NF-κB, the NLRP3 inflammasome, and mTOR. Therefore, ST is a protective agent against STZ-induced liver injury, pending further investigations to explore other mechanisms.

## Figures and Tables

**Figure 1 diseases-11-00184-f001:**
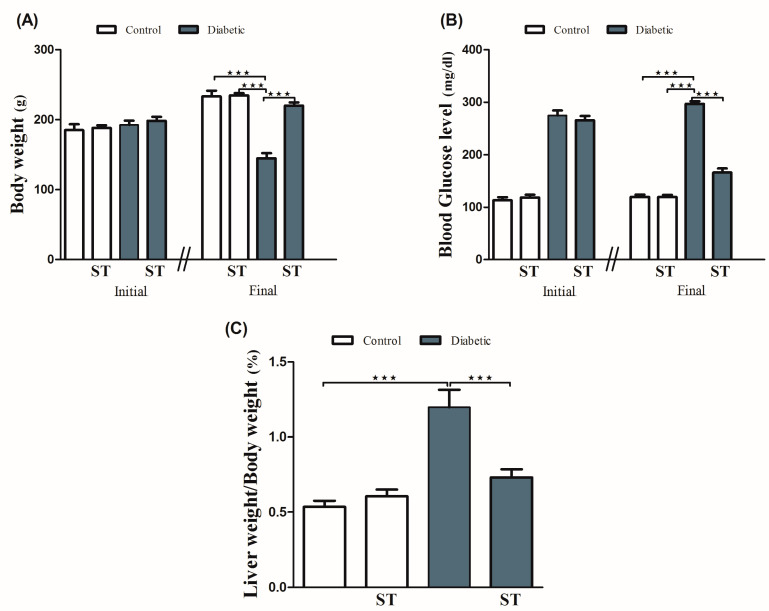
ST modulates the body weight (**A**), blood glucose (**B**), and liver/body weight ratio (**C**) in STZ-induced hepatic damage in diabetic rats. Diabetic and nondiabetic rats were either treated with physiological saline or ST (100 mg/kg) for 90 days. Body weight, blood glucose levels, and liver weight were measured at the end of the treatment period. ST, Sitagliptin; STZ, Streptozotocin. Data are mean ± SEM (n = 8). *** *p* < 0.001.

**Figure 2 diseases-11-00184-f002:**
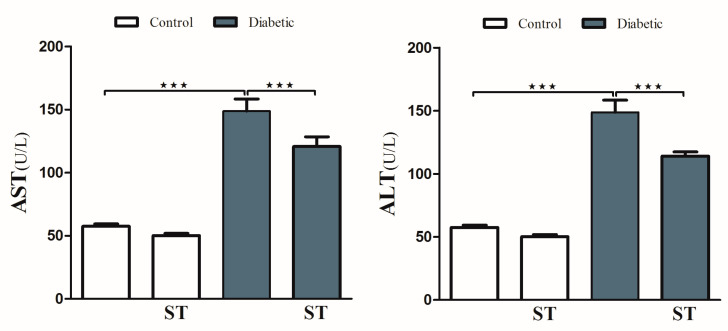
ST reduces the liver function enzymes (AST and ALT) in STZ-induced hepatic damage in diabetic rats. After treatment of animals either with vehicle control or ST (100 mg/kg) for 90 days, the liver enzymes ALT and AST were measured with biochemical kits. AST, aspartate aminotransferase; ALT, alanine aminotransferase; ST, Sitagliptin; STZ, Streptozotocin. Data are mean ± SEM, (n = 8). *** *p* < 0.001.

**Figure 3 diseases-11-00184-f003:**
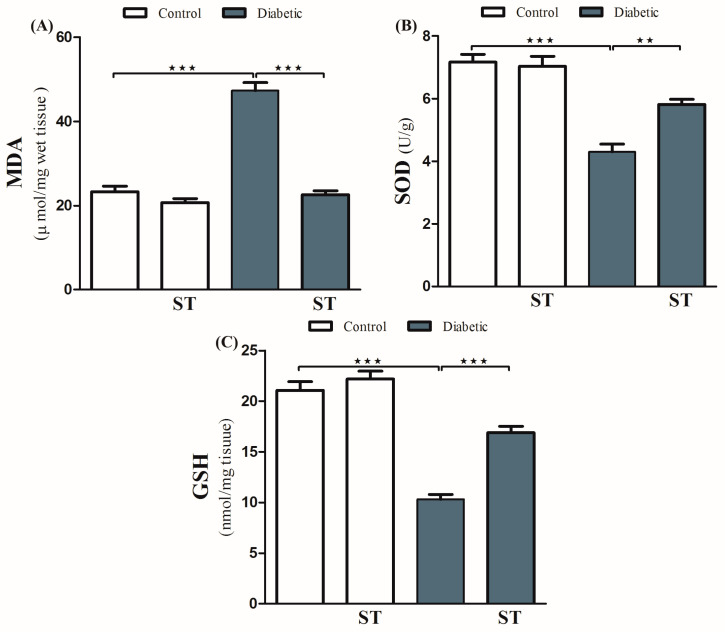
ST attenuates hepatic oxidative stress in STZ-induced diabetic rats. Treatment of rats with 100 mg/kg ST for 90 days decreases hepatic MDA (**A**) and increases SOD (**B**) and GSH (**C**) in diabetic model. MDA, malonaldehyde; SOD, superoxide dismutase; GSH, reduce glutathione; ST, Sitagliptin; STZ, Streptozotocin. Data are mean ± SEM, (n = 8). ** *p* < 0.01, *** *p* < 0.001.

**Figure 4 diseases-11-00184-f004:**
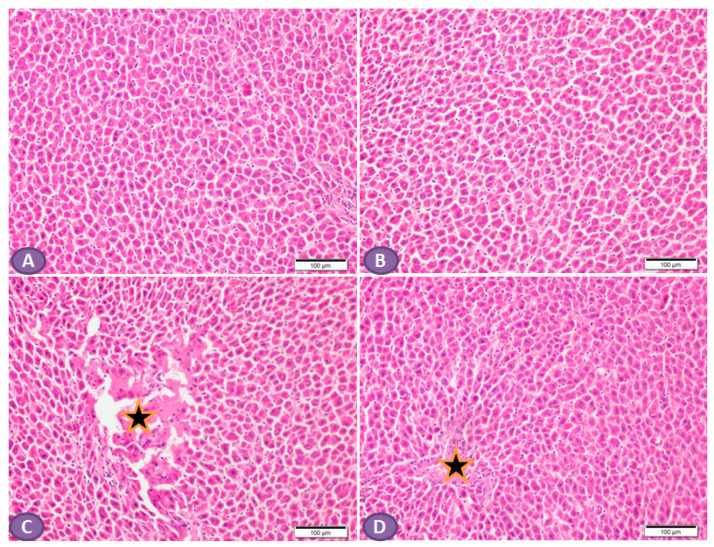
Photomicrograph of H&E-stained liver section, scale bar = 100 µm. (**A**,**B**) represent normal nondiabetic control liver and liver of rat who received ST, respectively, showing normal histological architecture of hepatic lobules and portal tracts. Hepatocytes and the hepatic blood sinusoid are in normal state. (**C**) represents a liver section from rat receiving STZ as a model of diabetes, showing multiple patches of hepatic degeneration with loss of group of hepatocytes (star). The surrounding tissues show inflammatory cellular infiltrations and hepatocyte cytoplasmic degeneration. Some hepatocytes show nuclear pyknosis. (**D**) represents liver from diabetic rat who received ST, showing marked decrease in inflammatory cellular infiltrations; most hepatocytes are apparently within normal cytoplasm and nuclei. There is recovery of the hepatic lobular architecture. The degenerated patches decreased in number and size (star). ST, Sitagliptin; STZ, Streptozotocin.

**Figure 5 diseases-11-00184-f005:**
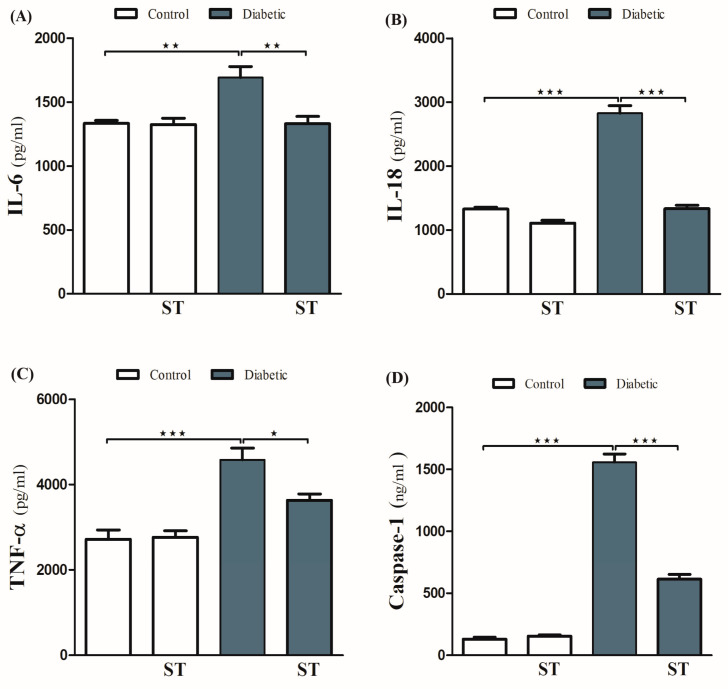
ST Downregulated hepatic inflammatory and apoptotic markers in STZ-diabetic rats. Treatment with ST decreases hepatic IL-6 (**A**), IL-18 (**B**), TNF-α (**C**), and Caspase-1 (**D**). ST; Sitagliptin, STZ; Streptozotocin. Data are mean ± SEM, (n = 8). * *p* < 0.05, ** *p* < 0.01, *** *p* < 0.001.

**Figure 6 diseases-11-00184-f006:**
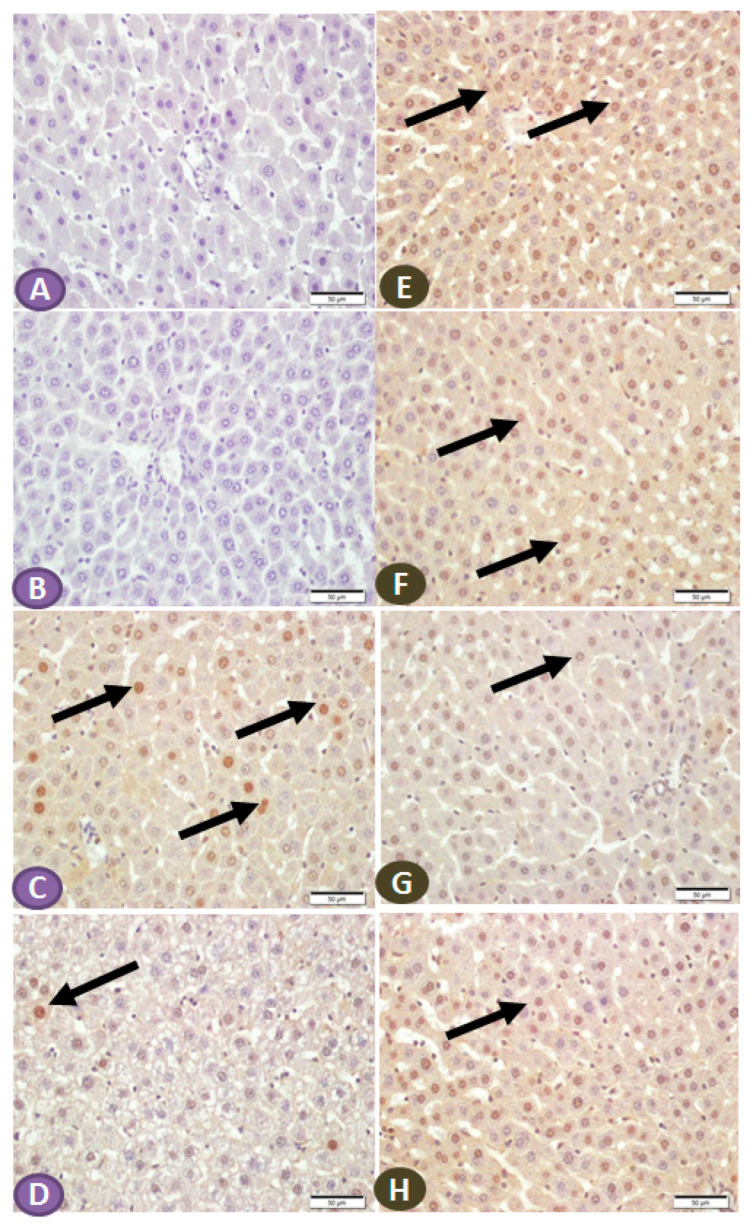
(**A**–**D**) Photomicrograph of NF-κB/p65 immune-stained liver sections, scale bar = 50 µm. (**A**,**B**) represent normal control liver and liver of rat that received ST, respectively, showing normal hepatocytes and absence of nuclear immune signal. (**C**) represents liver sections from diabetic rats showing many hepatocytes with strongly immune-stained nuclei (arrows), while section from rats treated with ST (**D**) shows few immune-stained hepatocytes nuclei (arrows). (**E**–**H**) Photomicrograph of IKB-α immune-stained liver sections, scale bar = 50 µm. (**E**,**F**) represent normal control liver and liver of rat received ST, respectively, showing normal positive strong immune reaction of hepatocyte nuclei. (**G**), liver section from diabetic rat showing few hepatocytes with immune-stained nuclei (arrow), while section from diabetic rat treated with ST (**H**) shows a marked increase in the number of hepatocytes with positively immune-stained nuclei (arrow). ST, Sitagliptin; STZ, Streptozotocin.

**Figure 7 diseases-11-00184-f007:**
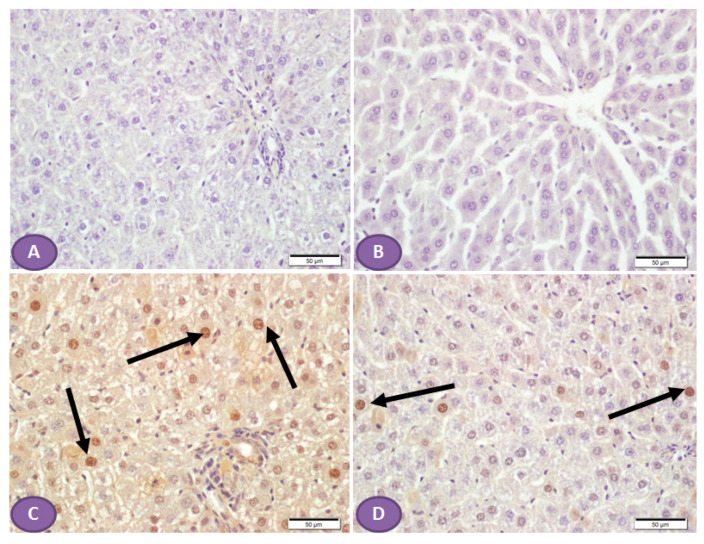
Photomicrograph of NLRP3 immune-stained liver sections, scale bar = 50 µm. (**A**,**B**) are of normal control liver and liver of rat who received ST, respectively, showing normal absence of nuclear immune reaction. (**C**) Liver section from diabetic rat shows many hepatocytes with immune stained nuclei (arrows), while section from ST treated rat (**D**) shows a marked decrease in the hepatocyte nuclear immune reaction (arrows). ST, Sitagliptin.

**Figure 8 diseases-11-00184-f008:**
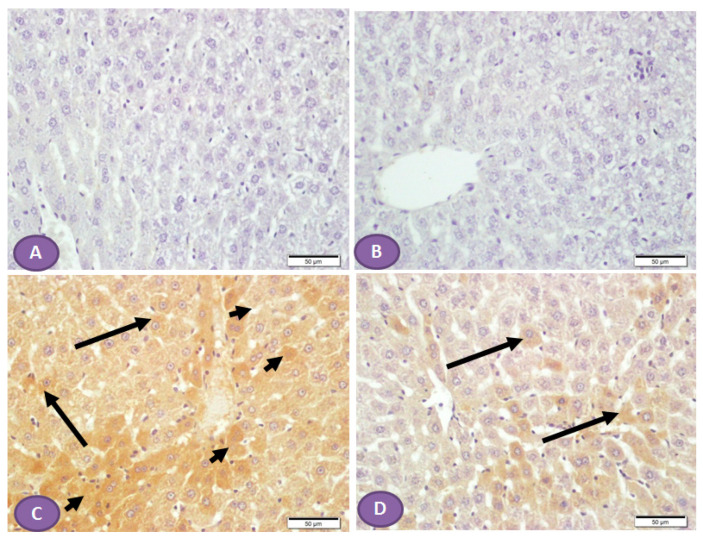
Photomicrograph of mTOR immune-stained liver sections, scale bar = 50 µm. (**A**,**B**) Photomicrograph of normal control liver and liver of rat who received ST, respectively, showing normal absence of immune reactivity, while (**C**) represents liver from diabetic rat, shows patches of hepatocytes with strong cytoplasmic immune reactivity (arrowheads); the rest of the hepatocytes show cytoplasm with weak immune reaction (arrows). (**D**) represents liver from diabetic rat that received ST showing small patches of hepatocytes with weak immune reaction (arrows) while the rest of hepatocytes have no immune reaction. ST, Sitagliptin.

**Table 1 diseases-11-00184-t001:** Effect of Sitagliptin on serum lipid profile (levels of total cholesterol (TC), triglycerides (TG), and high-density lipoprotein (HDL)) in control and diabetic rats.

Group	TC (mg/dL)	TG (mg/dL)	HDL (mg/dL)
Control	64.1 ± 1.53	66.1 ± 2.04	34.6 ± 1.04
ST	60.6 ± 1.74	63.5 ± 1.36	32.6 ± 1.96
STZ	205 ± 2.40 ***	181 ± 3.30 ***	18.8 ± 0.669 ***
STZ + ST	141 ± 4.74 ^###^	125 ± 2.71 ^###^	31.8 ± 0.632 ^###^

Data are expressed as mean ± SEM, n = 6, *** *p* < 0.001 versus Control and ^###^ *p* < 0.001 versus Diabetic.

## Data Availability

Data analyzed or generated during this study are included in this manuscript.

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
