# Peer review of "Protective Effects of Sitagliptin on Streptozotocin-Induced Hepatic Injury in Diabetic Rats: A Possible Mechanisms"

_diseases, 2023, doi:10.3390/diseases11040184_

Round 1

Reviewer 1 Report

Comments and Suggestions for Authors

Sitagliptin (ST) on diabetes-induced liver injury is investigated in current report. I like to give the following comments.

1.      ST is known as a dipeptidyl peptidase-4 (DPP4) inhibitor, hepatoprotective activity of ST associated with GLP-1 or not that shall be introduced in detail.

2.      In Line 82, DPPA4 seems an error. Same as King Saud University in line 118.

3.      Rats with blood glucose levels >200 mg/dL belonged to type-1 or type-2 diabetic model?

4.      The rats received ST (100 mg/kg) for 90 days needs the rationale and reliable background.

5.      Changes in DPP-4 activity by ST were ignored. Why?

6.      Results of autophagy lacked quantification.

7.      ST activated the AMPK/mTOR pathway in rat testes. How to link it to hepatic function?

8.      The treated dose of ST is important. How to distinguish the effect of ST on hyperglycemia via GLP-1 or pleiotropic effect in current report?

Comments on the Quality of English Language

It seems better to check through professional editing because errors were observed.

Author Response

Reviewer 1

First of all, I would like to extend my sincere thanks and gratitude for your precious time, we have revised the issues brought up by the reviewers, point-by-point response can be found below:

Sitagliptin (ST) on diabetes-induced liver injury is investigated in current report. I like to give the following comments.

  1. ST is known as a dipeptidyl peptidase-4 (DPP4) inhibitor, hepatoprotective activity of ST associated with GLP-1 or not that shall be introduced in detail.

Reply:

Thank you for your valuable comment, we modified the introduction section in the manuscripts and add more detail for the hepatoprotective effects for ST associated with GLP-1 that were highlighted line (84-93).

  1. In Line 82, DPPA4 seems an error. Same as King Saud University in line 118.

Reply: Done

  1. Rats with blood glucose levels >200 mg/dL belonged to type-1 or type-2 diabetic model?

     Reply:

The T2DM rat model confirmed by STZ-injection in previous recent studies report that the glucose level after 72 h STZ-injection >200 mg/dL belonged to type-2 diabetes.

For example:

1-    Ali et al., 2020, (Life Sciences 245, 117361). https://doi.org/10.1016/j.lfs.2020.117361

2-    Mridula Sharma et al., 2023, fundamental and clinical pharmacology, 4(37) 769-778https://doi.org/10.1111/fcp.12892                                           

  1. The rats received ST (100 mg/kg) for 90 days needs the rationale and reliable background.

Reply:

Dose selection of Sitagliptin as 100 mg/kg has been determined based on several previous studies that were conducted on rats. For instance, one research that involved studying the possible synergistic effect of Sitagliptin and Bromocriptine in reducing blood sugar in diabetic rat models employed this dose (Nileshraj et al., 2021). In addition, another study that was investigating the protective effect of Sitagliptin and whole-body γ-irradiation in diabetes-induced cardiac injury involved the administration of Sitagliptin in a dose of (100 mg/kg per day) for rats ( Mansour et al., 2021). Furthermore, a third study that was conducted on rats and to prove that chronic treatment with two DPP-4 inhibitors, Vildagliptin (20 mg/kg/day) or Sitagliptin (100 mg/kg/day), in these selected doses reduced myocardial infarction MI size (Hausenloy et al., 2013).

Ref.

Nileshraj, G., Swithraa, C., Sakthibalan, M. & Sawadkar, M. S. Study on Synergistic Effect of Bromocriptine and Sitagliptin in Streptozotocin-induced Diabetic Rats. J. Clin. Diagnostic Res. (2021) doi:10.7860/jcdr/2021/47643.14799.

Mansour SM, Aly S, Hassan SHM, Zaki HF. Protective effect of sitagliptin and whole-body γ-irradiation in diabetes-induced cardiac injury. Can J Physiol Pharmacol 2021;99:676-84

Hausenloy DJ, Whittington HJ, Wynne AM, Begum SS, Theodorou L, Riksen N, et al. Dipeptidyl peptidase-4 inhibitors and GLP-1 reduce myocardial infarct size in a glucose-dependent manner. Cardiovasc Diabetol 2013;12:154

  1. Changes in DPP-4 activity by ST were ignored. Why?

Reply:    

 Thank you for your valuable comment, we modified the introduction and discussion parts in the manuscripts and add more details about effects of ST on DPP-4 activity that were highlighted.

  1. Results of autophagy lacked quantification.

Reply:    

   Thank you for your valuable comment, all immunohistochemical figures in this manuscript have been extensively discussed.

  1. ST activated the AMPK/mTOR pathway in rat testes. How to link it to hepatic function?

Reply:    

     Yes, we mentioned the protective effects of ST in different organs partially through its modulatory mechanism on mTOR protein to confirm the implication of mTOR in the protective activity of ST.

  1. The treated dose of ST is important. How to distinguish the effect of ST on hyperglycemia via GLP-1 or pleiotropic effect in current report?

Reply:    

We sought to employ the study design that involves the administration of Sitagliptin in a dose of (100 mg/kg) to the rats for duration of 90 days to investigate the long-term effects and safety profile of this medication. That is because of the chronic nature of type 2 diabetes which often requires long-term pharmacological management. Thus, studying the effects of Sitagliptin over a 90-day period in rats probably allow us to simulate a chronic exposure situation, providing insights into the drug's sustained efficacy and potential side effects over time. This might be also feasible approach to translate our results into clinical practice. In addition, 90-day duration is long enough to examine various endpoints, such as changes in change in body weight, and liver weight/body weight, blood glucose levels, hepatic function, biomarkers analysis and potential histopathological alterations in tissues.

Best Regards

Reviewer 2 Report

Comments and Suggestions for Authors

The results are reported in a concise and easy-to-read manner, the content is legitimate, and I do not see any major changes.

However, there are a few areas of the discussion that could use some restructuring. First, have there been similar reports before? And if so, what new mechanisms have we identified in addition to those in the literature? If so, what is the new mechanism that our study adds to the literature? There are significantly fewer references to similar studies in our study. Also, Ghorpade DS,et al. Nature 2018;555:673-677 is very important literature and should definitely be added to the Introduction or Discussion.

Author Response

Reviewer 2

First of all, I would like to extend my sincere thanks and gratitude for your precious time, we have revised the issues brought up by the reviewers, point-by-point response can be found below:

  • However, there are a few areas of the discussion that could use some restructuring.

Reply:

Thank you for your valuable comment, we modified the discussion part in the manuscripts and add more studies that were highlighted.

  • First, have there been similar reports before? And if so, what new mechanisms have we identified in addition to those in the literature? If so, what is the new mechanism that our study adds to the literature?  There are significantly fewer references to similar studies in our study.

Reply:

Some studies showed the effect of ST on antioxidant and anti-inflammatory pathways. Each added mechanism was highlighted in the text,

Lines (475)

Besides the liver, ST can attenuate of STZ-induced oxidative stress and glomerular lesions by decreasing DPP4 protein levels and upregulating GLP-1 in renal tissue (Marques  et al., 2019)

Lines (522)

ST showed hepatoprotective effects against thioacetamide induced liver injury by downregulating NLPR3 inflammasome production and TLR4 and NK-KB protein expression (El-Kashef et al., 2019). In addition, ST was effective against inflammation, fibrosis and tubulointerstitial injury in doxorubicin induced nephropathy. The gene expressions of NLRP3, caspase-1, and IL-1β were elevated during doxorubicin use but reversed with ST treatment (Jo et al., 2018). The inhibitory effect of ST on these proteins can be mediated through protein kinase C inhibition (Dai et al., 2014). However, to the best of our knowledge there no study reported the effect of ST on NLRP3, caspase-1, and IL-1β in liver after STZ induced DM which emphasized the importance of this study.  

Lines (547)

In polycystic ovary syndrome (PCOS) rat models, ST inhibited autophagy and inflammation by inhibiting the PI3K/AKT/mTOR and TLR4/NF-κB pathways, respectively (Ren et al., 2023)

This work showed for the first time the hepatoprotective effects of ST against DM induced via STZ were mediated through controlling the expression of NF-κB, NLRP3 inflammasome, and mTOR.

  • Also, Ghorpade DS,et al. Nature 2018;555:673-677 is very important literature and should definitely be added to the Introduction or discussion

Reply:

Thank you, done this reference is added and highlighted in the introduction

Best Regards
